**Data Availability Statement:** All relevant data are within the paper.

**Funding:** The author(s) received no specific funding for this work.

# Women's perception about the discovery of breast cancer amid the covid-19 pandemic

**Simone Meira Carvalho**[1]*, **Camilla de Abrahão Andrade**[2], **Mariana Barbosa Leite Sérgio Ferreira**[3], **Karine Soriana Silva de Souza**[3], **Fabiane Rossi dos Santos Grincenkov**[3]

**1** Department of Physical Therapy for Seniors, Adults and Mother-Child, School of Physical Therapy, Universidade Federal de Juiz de Fora (Federal University of Juiz de Fora), Juiz de Fora, Minas Gerais, Brazil, **2** School of Physical Therapy, Federal University of Juiz de Fora, Juiz de Fora, Minas Gerais, Brazil, **3** Department of Psychology, Institute of Human Sciences, Federal University of Juiz de Fora, Juiz de Fora, Minas Gerais, Brazil

\* simeiracarvalho@hotmail.com

## Abstract

### Background

Breast cancer is considered a health problem at a worldwide level. In Brazil, the South and Southeast regions have the highest mortality rates. Understanding how they dealt with the diagnostic of a stigmatized disease amid the COVID-19 pandemic and its potential repercussions, may enable healthcare professionals to of life. Thus, this study is aimed at understanding the perception of women about the discovery of breast cancer and the impact of the disease on their lives.

### Methods

A qualitative study, with the participation of forty women with breast cancer, under chemotherapy treatment. It was performed in a hospital specialized in oncology, in Juiz de Fora, Brazil, in 2020 and 2021. Data collection was carried out with semi-structured interviews, which were analyzed with Bardin Content Analysis.

### Results

Based on the central theme "Discovery of the disease", these categories were developed: "Discovery" and "Impact of the disease". A large part of women noticed a change in the breast, even before routine checks. Upon the impact of cancer diagnosis, negative feelings arise, then going through a process of acceptance and coping. Some barriers were faced due to the COVID-19 pandemic, which caused delays in the diagnostic and impact caused by social isolation. Family, friends, and healthcare professionals integrated an important supporting network in order to help coping with the disease.

### Conclusion

The consequences of a breast cancer diagnosis can be devastating. It is necessary that healthcare professionals know and embrace the feelings, beliefs, and values as a part of the aspects related to health. Valuing the supporting network of women suffering from the

**Competing interests:** The authors have declared that no competing interests exist.

disease may favor the process of accepting and coping with the neoplasm. The COVID-19 pandemic is highlighted as an obstacle to be overcome specially when it comes to diagnostic assistance and availability of a support network. In that sense, it is worth mentioning the importance of a healthcare team able to offer full assistance, with quality. The need of further studies to determine the impact of the pandemic in the long run.

## 1. Introduction

Breast cancer is considered a significant health challenge at a worldwide level due to its high incidence and high mortality rate for women. In Brazil, 66,280 new cases were found for each year of the three-year period of this study, which ranged from 2020 to 2022, according to data from the Brazilian National Cancer Institute (INCA). Such neoplasm is the main cause of death among women, with the highest mortality rates occurring in the Southeast and South regions [1]. It is also the type of cancer that scares women the most, due to its negative impact on overall quality of life [2, 3]. In many cases, breast cancer is detected by the patients themselves, through a self-exam, and subsequently confirmed with a mammogram screening and a biopsy [4, 5].

The discovery of breast cancer triggers physical as well as emotional effects, degrading the quality of life of those suffering from it [6]. Upon receiving the news of the cancer, feelings of helplessness are overwhelming for women, since this diagnostic will prove devastating for their lives [3, 7]. Due to this severe psychological shock, there is a need of support from relatives and friends, as well as of assistance from qualified professionals able to meet the emerging demands at this crucial moment [2, 3, 8].

Diagnostic and treatment in early stages of the disease increase the chances of success in the treatment, evincing the need of early identification of the illness to approach it with less aggressive therapies and with a more favorable prognosis [9, 10]. However, an unexpected event interfered in that process. Due to the outbreak of the coronavirus SARS-CoV-2 (COVID-19) on the world stage, a delay in the diagnostic and start of treatment of breast cancer was identified in 2020 [11, 12], causing worst prognosis and higher chances of mortality [13]. The temporary shutdown of several sectors, including those related to healthcare, and the need of social isolation dramatically reduced the number of mammogram screenings and other diagnostic tests, especially in the second quarter of that year [13], which raised the concern of healthcare teams and patients with cancer regarding the prognosis, due to the postponement or changes in the oncological treatment and medical examinations [14, 15].

Thus, questions have risen about the discovery of breast cancer in women and the implications of the diagnostic of such pathology. According to a systematic review carried out by Arab, Correia, Demonico, Vilarino and Andrade [2], scientific production in Brazil on the implications of cancer in mental health is still modest, especially when compared to the scientific production on breast cancer using a quantitative sampling. Another systematic review, published in 2022, shows that there are a few studies about it, but it highlights the benefit of using this topic, considering the high rates of breast cancer [8].

Several authors emphasize that the discovery of the pathology is traumatic for women, triggering confusing feelings and deeply changing their lives [8, 14, 15]. Regardless of family support, women become vulnerable, frightened and they often lose faith in finding a proper treatment to fight the disease [7]. Moreover, the COVID-19 pandemic seemed to have interfered in the investigation process. In addition to the suffering caused by the illness, it was possible to identify an unfavorable repercussion in both emotional welfare and quality of life [16].

In face of such outlook, the purpose of this study is to understand how women have dealt with the news of the diagnostic of a stigmatized disease, and their perception of cancer as a 'death sentence' amid the COVID-19 pandemic. Therefore, this study is relevant to enable healthcare professionals and the academic community to widen their knowledge on the aspects involving breast cancer, especially in this unique moment of the pandemic and its potential repercussions. Understanding the factors that involve the process of the illness may collaborate with the improvement of quality care to women suffering from cancer. Healthcare with an integral view must encompass the demands and the needs of people with cancer beyond focusing on the disease. Based on these assumptions, this study aims at understanding the perception of women about the discovery of breast cancer and the impact of the disease on their lives.

## 2. Methods

### 2.1. Project

This is an exploratory, cross-sectional descriptive study, with qualitative approach and an intentional sample comprised of 40 women diagnosed with breast cancer, undergoing chemotherapy treatment. The study was developed in an oncology-specialized hospital, located in the city of Juiz de Fora, Brazil, which assists the local population as well as people from surrounding cities. It was carried out from January to March 2020, and from October, 2020 to February, 2021. The participants responded to an individual semi-structured interview. Qualitative approach was applied in the interpretation of the content of speeches, as it worked with a realm of motivations, meanings and attitudes from the subjectivity area. As a methodological guideline, Content Analysis was chosen, aiming at understanding the experience of the phenomenon [17], i.e., the discovery of breast cancer and its impact on the life of women. The guidelines of *Consolidated Criteria for Reporting Qualitative Research* (COREQ) qualitative research reporting guide were used to describe the data in this article.

### 2.2. Sample profile

The sample of the study was intentional, composed of women in a situation of similarity regarding the diagnostic of breast cancer, assisted by the Brazilian Public Health System (SUS) in the chemotherapy sector of the referred hospital. The diversity of the sample is characterized by varied ages and by the participants' place of origin, which are cities belonging to the Zona da Mata macroregion, in Minas Gerais, which are assisted in Juiz de Fora. For the purpose of being included in this study, women submitted to adjuvant or neoadjuvant chemotherapy, with or without breast surgery and aged over 18 years-old were approached. Women with local relapse or metastasis, time from the surgery longer than 18 months, education below 4th year of primary school or any cognitive impairment that would hinder the ability to understand the questions of the interview were not included.

### 2.3. Instruments

The semi-structured interview addressed social-demographic and clinical data, as well as questions on the discovery of cancer, treatments carried out and their effects, guidance, and self-care as main topics. The main researcher designed the questions of the interview, considering the literature in that area [18–20] and the experience with qualitative research in the field of women's health and breast cancer, as a physiotherapist, faculty, master in collective health and doctoral candidate in psychology. Before being applied, the interview script was reviewed by professional researchers from different fields of healthcare and was tested with women in a similar situation to those of the sample, who did not meet all requirements. A pilot study that

enabled the consolidation of the script as an instrument to achieve the objectives of this study was also carried out.

## 2.4. Selection of participants

There was no previous acquaintance or bond between the researcher and the interviewees. Therefore, participants' enrollment started upon contact with the nurse in charge at the chemotherapy clinic of the hospital, who revealed which women had breast cancer. The approach happened during the application of chemotherapy, in a direct conversation with the patient, with no companion. At that moment, the main researcher would introduce themselves and start a conversation, checking if the patient would meet the eligibility criteria. Upon meeting such criteria, the patient would be invited to participate in the study, and its objectives would be explained. In the invitation, the Free and Informed Consent Form (FICF) was presented in writing to each of the invitees who agreed to participate in the research. After reading and clarifying doubts regarding the informed consent, the women were asked to sign the form, and a printed and signed copy remained with the researcher and another with the participant. From 80 women approached, 40 did not enter the research for several reasons, and the selection was completed with 40 participants, as shown in the flowchart (Fig 1).

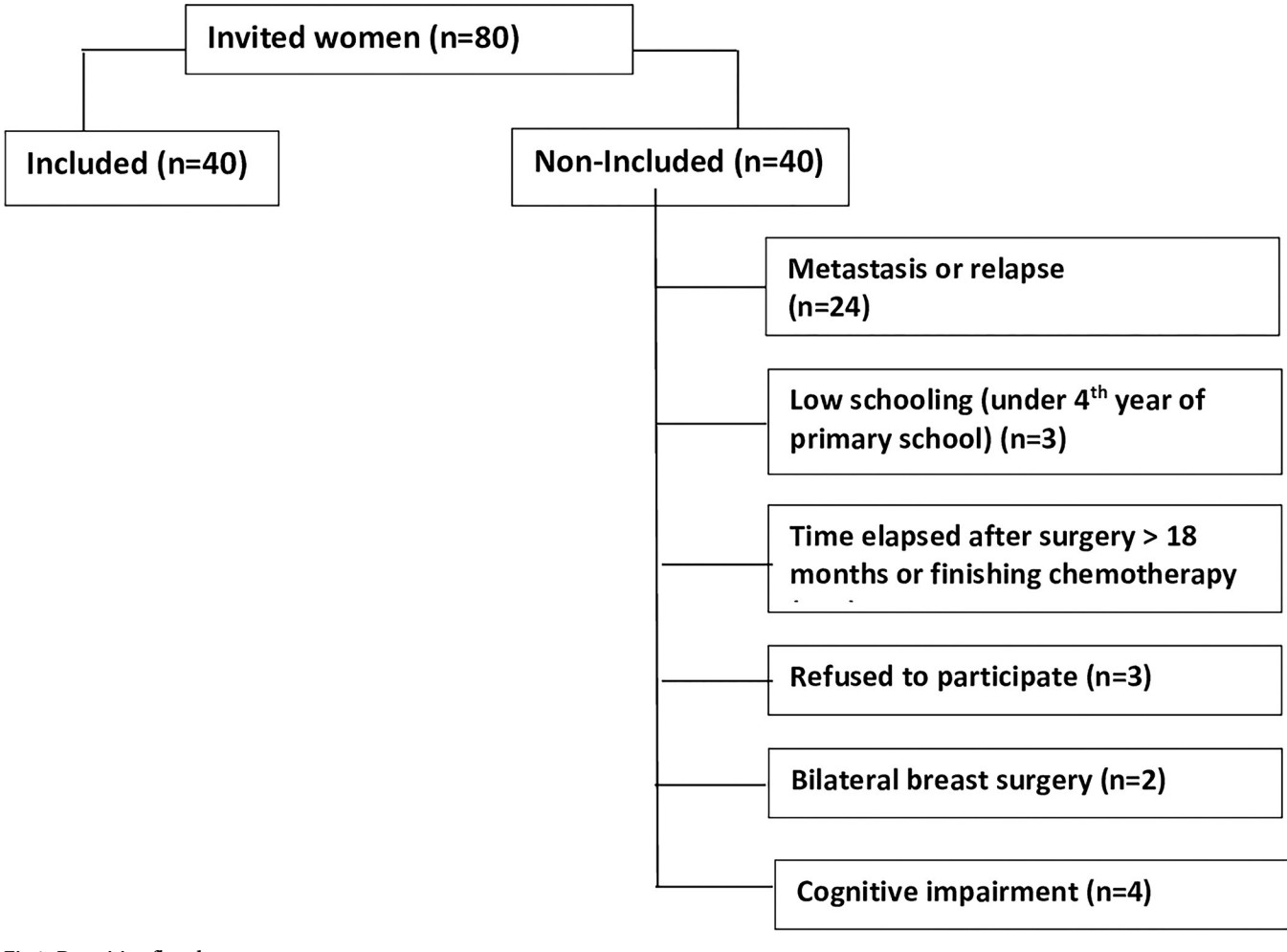

**Fig 1. Recruiting flowchart.**

## 2.5. Data collection

As the ICF was signed, data was collected through an individual interview, following a semi-structured script that started with questions that encouraged women to talk about the discovery of breast cancer. The interview was carried out in person, at the moment of chemotherapy application. The interviews lasted in average 24 minutes and were recorded and transcribed in full. The transcriptions were checked by the main researcher, to guarantee reliability of information. Some field notes were recorded during the interview, to capture non-verbal expressions that could enrich the perception of the interviewee about the topics discussed. Such notes were used as a supplement to the analysis material. The medical records of the participants were accessed aiming at completing the information on social-demographic and clinical data of the interviewees. Partial data collection was finished with 40 interviews, observing the saturation criteria, with recurrence of accounts, and allowing to achieve some objectives of the research. This article is a cross-sectional cut-off from a doctorate research that is still in progress. Data collection was performed from January 2020 to February 2021, being interrupted between March and October 2020, due to the COVID-19 pandemic.

## 2.6. Data analysis

For qualitative approach of the interviews, Content Analysis with thematic framework was adopted, as proposed by Bardin [17]. The systematized procedures were performed manually by the main researcher, in three steps. Transcription readings and preparation of indicators were carried out during the pre-analysis step, respecting the completeness criteria in reading, and understanding, representativeness of speech, homogeneity on the topics and pertinence, in order to achieve the purpose of the research. The main recording units for material encoding and analysis category development were assigned in the material exploration step, observing the exclusivity of the assigned elements. The topics and their respective categories emerged during information selection and condensation. In the final step, treatment of the findings, inferential interpretation of the listed categories was performed. The two final steps were performed upon discussion with the remaining authors, in order to achieve reflectivity of the findings and thoroughness in the analysis process. The analysis of the content of the explored material gave rise to several thematic axes and categories, regarding the discovery of breast cancer, the beliefs about the disease, the perceptions regarding treatments and self-care during the treatment period. It should be clarified that, given the extent and density of the findings, this report specifically presented the axis "Discovery of the disease" and their respective categories, which are detailed in the findings of this article. The categories are exemplified by participants' citations, who were identified solely by the letter E followed by a number, ensuring the confidentiality of their identity. For exposure of sociodemographic and clinical data, the variables were analyzed and presented as absolute and relative frequency.

## 2.7. Ethical standards

The project was approved by the Research Ethics Committee of the Federal University of Juiz de Fora (opinion no. 3.649.430, on 10/18/2019). This article is a cross-sectional cut-off of partial data from a research that is in progress. The researchers sought to respect the confidentiality and all other ethical aspects in the study process, according to the resolution no. 466, issued on December 12th, 2012, by the National Health Council/Ministry of Health. Before data collection, the researcher detailed the objectives of the research to the clinical management of the hospital in a letter of support. Therefore, consent was given by the hospital to perform the study. During data collection, at the time of reading the FICF, the participants were informed that the sole purpose of the information provided by them is to achieve the objectives of the

research. In order to check the secrecy of their identities, it was decided to index the participants' speeches by the letter E (from the word interview in Portuguese—*entrevistada*), followed by an Arabic number, and the names mentioned by them were replaced with other random letters.

# 3. Results

## 3.1. Social-demographic profile of the participants

The social-demographic data of the participants of the study are described in Table 1 below, where it was observed a prevalence of brown skin (42.5%), with ages ranging from 31 to 77

**Table 1. Social-demographic characteristics.**

| Variable | | N(%) |
|---|---|---|
| **Age** | | |
| | < 50 years-old | 10 (25.0) |
| | ≥ 50 years-old | 30 (75.0) |
| **Skin color** | | |
| | White | 15 (37.5) |
| | Brown | 17 (42.5) |
| | Black | 8 (20.0) |
| **Marital status** | | |
| | Single | 7 (17.5) |
| | Married | 22 (55.0) |
| | Divorced | 5 (12.5) |
| | Widow | 6 (15.0) |
| **Profession** | | |
| | Household assistant | 4 (10.0) |
| | Self-employed | 2 (5.0) |
| | Retired | 11 (27.5) |
| | Under health leave | 20 (50.0) |
| | Homemaker | 3 (7.5) |
| **Family income** | | |
| | 1–3 national minimum wages | 34 (85.0) |
| | >3–5 national minimum wages | 6 (15.0) |
| **School level** | | |
| | Complete primary school | 2 (5.0) |
| | Incomplete primary school | 14 (35.0) |
| | Complete secondary school | 14 (35.0) |
| | Incomplete secondary school | 6 (15.0) |
| | College degree | 3 (7.5) |
| | Undergraduate (incomplete college course) | 1 (2.5) |
| **Religion** | | |
| | Catholic | 29 (72.5) |
| | Evangelical | 7 (17.5) |
| | Other | 4 (10.0) |
| **Religious practice** | | |
| | Practicing | 32 (80.0) |
| | Non-practicing | 2 (5.0) |
| | Not very active | 6 (15.0) |

years-old. Regarding school level, incomplete primary school education (35.0%) complete secondary school education (35.0%) were prevalent. Part of the women (37.5%) were married. Half of them (50%) were on health leave and the majority (85.0%) had a family income between 1 and 3 national minimum ages. Catholic religion was prevalent (72.5%), with 80.0% of the women claiming to practice Catholicism. Over half (57.7%) of the participants were from Juiz de Fora, and the remaining (42.5%) lived in surrounding cities (Table 1).

### 3.2. Clinical profile of participants

Regarding the clinical profile, most (60%) of the participants presented comorbidities, among which are Systemic Arterial Hypertension, Diabetes, and obesity. The majority were non-smokers (62.5%) and non-alcoholics (70.0%). Regarding local therapies, most of them had already undergone partial breast removal surgery (65.0%) and 62.5% had not undergone radiotherapy until the time of the interview. Regarding systemic treatment, 40.0% were on chemotherapy alone and 25.0% were on neoadjuvant systemic therapy, 45.0% were also on Target Therapy and the others were on these two therapies associated with hormone therapy (10.0%) (Table 2).

**Table 2. Clinical characteristics.**

| Variable | | N(%) |
|---|---|---|
| **Comorbidities** | | |
| | Yes | 24(60) |
| | No | 16(40) |
| **Smoking** | | |
| | Yes | 6(15,0) |
| | No | 25(62,5) |
| | Former smoker | 9(22,5) |
| **Alcoholism** | | |
| | Yes | 2(5,00) |
| | No | 28(70,0) |
| | Socially/Sporadically | 10(25,0) |
| **Surgery** | | |
| | Mastectomy (radical) | 4(10,0) |
| | Sectorectomy (conservative) | 26(65,0) |
| | Does not apply | 10(25,0) |
| **Radiotherapy** | | |
| | Yes | 5(12,5) |
| | No | 25(62,5) |
| | Does not apply | 10(25,0) |
| **Systemic therapies** | | |
| | Chemotherapy | 16(40,0) |
| | Target therapy | 0 |
| | Hormone therapy | 0 |
| | Chemotherapy + Hormone Therapy | 2(5,0) |
| | Chemotherapy + Target Therapy | 18(45,0) |
| | Chemotherapy + Hormone Therapy + Target Therapy | 4(10,0) |
| **Neoadjuvant therapy** | | |
| | Yes | 10(25,0) |
| | No | 30(75,0) |

Regarding the discovery of the disease, 21 women (52.5%) noticed a breast change by chance; 11 (27.5%) discovered it with a self-check and the others (20%) discovered it with a control mammogram. Still regarding the identification of cancer, the study considered the steps of discovery of breast changes, investigation of the disease and final diagnostic of cancer, which occurred in different moments in relation the COVID-19 pandemic. Half (50%) of the participants had gone through the three steps before the pandemic, five women (12.5%) had noticed breast changes before the pandemic but received the final diagnostic during the pandemic. The remaining participants (37.5%) went through the three steps during the pandemic.

## 3.3. Interview analysis

Analysis of the content of the explored material enabled record units to emerge. The units featured during the analysis were grouped and resulted in several central themes and their respective categories. This article approaches the theme "Discovering the Disease".

## 3.4. Discovering the disease

This theme approaches the various manners the disease is discovered, either due to breast changes noticed by women themselves or by routine control tests. It also explains the repercussion of the diagnosis in the life of the participants and the beliefs involving the disease. The analysis led to the theme categories: "the discovery" and "impact of the disease", detailed below.

**3.4.1. The discovery.**   In most cases, women themselves have found some breast change, usually reported as a lump or nodule. In many cases, the discovery happened by chance, and in part of them it was during the breast self-exam. The discovery situations varied from palpating the breast during a shower to an observation on the mirror.

*I was taking a shower, passed my hand on my armpit and felt the lump, I also felt a difference in my nipple, it was parched and hard. (E21)*

*I was in the shower, and I noticed that my right breast was different from the left, I went to my bedroom, and I looked on the mirror and I saw it. (E22)*

However, the perception of changes not always have led women to immediately look for healthcare services, possibly due to a lack of understanding the signs and symptoms of the disease.

*I have learned how to do the self-exam with my gynecologist. Then, in another day, in the shower, I did the self-exam and I felt the nodule, but I forgot about it. I just got scared when (. . .) I pressed my nipple, and some liquid came out. (E31)*

Some of the participants of the study, despite already performing control exams for the disease, either due to aging, as indicated by the Ministry of Health (MH), or due to cases of breast neoplasm in the family, perceived a change in the breast region by chance or while performing the breast self-exam.

*When I did the mammogram, there was nothing wrong. I used to do routine exams, every year. [. . .] One day, in the shower, I felt it. I had done the mammogram in November and discovered it in March. (E9)*

*I do a mammogram every year. Then, before the time had come to do the next one, (. . .) I felt the lump. (. . .) Then, I searched for help. It was already in an advanced stage but thank God I could solve it. (E19)*

*I discovered it with routine exams, mammogram (. . .), last October [2020] when a change appeared. (. . .) I was aware of other cases in my family* [of breast cancer]. *(E37)*

Despite this scene, performing exams with the purpose of early detection of breast cancer has proven to be effective in revealing the disease. From such exams, mammogram stands out as an imaging test capable of identifying lesions, nodules or asymmetries that indicate the development of a neoplasm.

*It was through a mammogram. It was dr. N [physician] who had requested it. She has detected it, as I had not had pain and had not felt anything. (E24)*

*I discovered it with routine exams, mammogram, in October last year [2020] when a change appeared. (E37)*

Due to the interruption of activities in several sectors of the society, known as "lockdown", the COVID-19 pandemic caused a delay in performing control exams for diseases, including cases of neoplasm.

*I have always done mammogram and ultrasound exams and took them to the doctor. [. . .], I would have to do it this year* [2020], *in February [. . .], but with the pandemic I could not, as everything was closed. By June* [2020], *my breast got red. Then I went to the doctor in the Community Health Center, and she requested an ultrasound. It was when it showed I had a nodule. (E25)*

**3.4.2. Impact of the disease.** The diagnostic of breast cancer triggers several emotions, according to the interpretation of each woman, as such disease carries the stigma of finitude. In this study, the reactions and feelings reported by the participants were scare, sadness, despair, and concern. Even though some had witnessed cases of cancer in the family or with close friends, surprise was present in the participants' reports.

*I got really scared, as I got a lump. (E16)*

*When the doctor told me what I had, what she really detected, I cried a lot. (E22)*

*(. . .) I was really terrified. I confess I felt so desperate. Because you see people around, in my house I had also seen my uncles [with cancer]. But when it happens to you, the situation is quite different. (E31)*

In face of so many negative feelings and so much insecurity, in the perspective of women, they feel completely lost. Which means they were facing the uncertainties of life. Such view triggered several reactions, causing some to choose to be alone when they received the diagnosis, in a movement of isolation, to cope with the impact of being with a disease that brings the possibility of death.

*At the moment, I remained strong. But when I left the office, it felt I got no ground under my feet. . . (crying). (E23)*

*The day I left the office and she told me what it was, I cried a lot. I felt lost, I could not see any-thing, I just wanted to be alone. I did not call anyone, I did not call my family, I did not even call my boyfriend. It was me and God. (E31)*

Support from healthcare professionals is highlighted as an important factor for the process of coping with and accepting the diagnosis. In that sense, assistance from a psychology profes-sional was key to assimilate the circumstance of being ill.

*I had two diagnostics, one in the breast and one in the armpit. I was quite nervous with the sit-uation, and then he [physician] sent me to the psychologist. (E17)*

*I was already feeling the nodule, so I realized it was going to be something more severe. At first, it was very complicated for me receiving the diagnostic. (E33)*

*I was a bit terrified but thank God I put something in my mind: "if it is for my well-being, let's treat it". [...] even the doctors themselves reassure me. I got a bit upset, but I did not feel so bad. (E14)*

The effort to accept a diagnostic related to the possibility of death goes through the relation-ship with relatives. In face of extreme situations, family is a motivating factor in coping with the diagnostic.

*At first, I felt very scared about everything. But we slowly assimilate a bit more and see that there is always a solution. [...] I have three grandchildren. Then, this gives me more strength. (E13)*

They simultaneously need to find strength to tell their relatives about it, knowing they would be dealing with barriers related to beliefs about cancer that could emotionally destabi-lize the family and change their interpersonal relations.

*(...) my mother is very close to me. On the day I found out, I did not tell my mother, [...] my sister went out to tell her, because I did not have the courage to tell her myself. (E28)*

*When it happens to other people, you say: "everything will be fine, you will see this through". Now, when it happens to you, you have to tell this to yourself. (...) I was telling her [grand-mother] little by little. (...) And I had to find strength I did not know I had in me, to tell my family (...). Because I know it would disrupt them with all that. (E31)*

Notwithstanding the need of support from the family at the moment of the diagnosis, the pandemic was a hindering factor for closer contact, triggering a feeling of helplessness even greater due to the need of social distancing demanded by the health situation.

*I lost my father last year, which was already too much for me. My father died of a cardiac arrest and (...) he also got Covid-19. (...) I got it too later. It was a moment when I needed a hug the most and I could not receive it due to Covid-19. (...) then I got the cancer diagnostic. (E37)*

## 4. Discussion

This study revealed that a significant number of women had identified breast changes by chance (52.5%) or at the moment of the breast self-exam (27.5%), confirming the findings of another research [3–5]. The remaining (20%) discovered the cancer through a mammogram.

The study was split into two categories, the first being related to the discovery, including the self-exam and the imaging tests indicated by the MH [10]. Although the self-exam is not currently proposed by that governmental body as an instrument for early diagnostic, the breast exam was reported by several participants as the manner by which they had found the first sign of cancer, the breast nodule [3]. Findings by Azriful et al. [4] and Costa et al. [5] suggest that the technique of self-detection of cancer symptoms helps to find the problem. The breast self-exam practice indicates greater perception of susceptibility to cancer, which is key to seek medical assistance and immediate treatment. Although in this research several women had noticed a breast change during the self-exam, such findings usually happen in a more advanced phase of tumor growth [9, 10]. Therefore, the self-exam does not downplay the importance of diagnostic exams indicated by the MH for early detection of the disease, such as the mammogram, as imaging exams enable better prognostic, with less aggressive interventions [10, 21].

This study also shows that, despite perceiving changes in the breast, some women did not pay attention to the possibility that it could be a neoplasm, which caused some delay in the diagnostic of the disease. According to Tesfaw, Alebachew and Tiruneh [22], low perception of the susceptibility to the disease results in lower likelihood to look for assistance in a health-care center. Understanding of the disease process is mediated by cultural, economic, and social aspects, especially in the case of cancer [3, 22]. Other factors may also interfere with the search for experts to keep doing exams that detect breast cancer. Some authors mention the lack of knowledge about risk factors and the lack of information on the signs and symptoms of breast neoplasm as prevalent factors that delay the diagnostic and have major emotional impact [2, 20, 22, 23].

Other factors impacted issues related to cancer tracking and diagnostic, due to the reduced offer of services and postponed medical exams, as was the case of the pandemic. Studies show that such situation may lead to a late diagnostic, with increased morbimortality risk, in addition to enhanced feelings of anxiety and depression [14, 15, 25]. Delay in several steps, from routine medical appointments and follow-up to diagnostic exams and cancer treatment occurred due to the need of social isolation and temporary shutdown of healthcare services, especially in the first months of the pandemic [11–14]. As time went by, healthcare services have adapted to the COVID-19 situation, developing security protocols against the disease in order to resume the operation of health services at normal levels. In that sense, it was considered that the risk of contamination by coronavirus is lower than the benefits of tracking cancer [13]. It is worth remembering that only 50% of the interviewees received a definitive diagnosis of breast cancer during the pandemic.

In the second analysis category, referred to as "Impact of the disease", a range of feelings was reported as experienced by women when they had the diagnostic confirmed, from fear and helplessness to acceptance. It was observed in this study that one of the most common feelings experienced by them is fear, which starts at the moment the disease is discovered and remains during the effects of the treatment, confirming the findings of other researchers [21, 23, 24]. In face of such negative feelings and insecurity regarding the future, many women withdrew, even if temporarily, from their families and other social relationships. This period is part of the process of adjustment and acceptance of disease [2, 21, 24]. In this situation, living with an oncological disease considered incurable by many people, brings one closer to the perception of the fragility of life, triggering feelings of despair and fear of leaving the family [21]. Add to this equation the feelings of concern and dismay for the pain of dear ones in face of the diagnostic, which was found in this study and mentioned in some research [3, 21, 23].

Even though the family suffers with the effects of the impact of breast cancer, support from relatives and friends contributes significantly to the process of acceptance and coping [5, 16],

as well as to the recovery of ill women and to the therapies [19, 20]. Thus, Martins et al. [20] emphasize the benefits of embracing and guiding relatives and people who have social interactions with the patient in order to increase their support. Research show that the supporting network composed of relatives, friends, healthcare professionals, faith in God and religious practices reduces anxiety, depression and contribute to fight the negative impact of cancer [5, 20]. Other studies show the need of psychological assistance to shelter the emotional pain of women in face of the disease, as this professional approach enables greater adhesion to treatments [6, 23, 24].

According to Massicote, Ivers and Savard [15], the pandemic had a negative impact on the quality of life of women with breast cancer, who already feel overwhelmed by that disease. In this period of health crisis, the emotional repercussions of breast cancer involve greater concern with the disease and health in general [14]. Although cancer may cause changes in social relations, the feeling of loneliness in the referred population was reinforced by the social isolation arising from the restrictions resulting from the pandemic. However, the same feeling was identified in a control group of women who were not suffering from cancer, in the research by Rentscher et al. [25]. In addition to loneliness, social isolation was related to increased anxiety, stress, and depression [25]. Additionally, to the barriers deriving from coronavirus, Azriful et al. [4] advocate that sadness, fear and depression are hindrances to overcome in the recovery of women with cancer, both in the period of its detection and during the treatment. According to a few authors, patients with breast cancer have been more susceptible to develop psychological issues in the context posed by the pandemic, a fact that emphasizes the need of social support to such individuals [8, 4, 15].

This study showed that coping with the diagnostic is part of a process that starts with a period of turmoil, goes through a period of adaptation, until reaching a period of serenity. It is important that the healthcare team is prepared to embrace the woman with her fears, insecurities, anxieties, changes in life habits and body image, considering all aspects, physical, psychological, cultural, and social, in order to improve her quality of life [2, 23, 24]. As the support from family and friends is essential for a better quality of life and adherence to the treatment, encouraging social interactions is an interesting strategy to strengthen bonds and coping with cancer [4, 24]. Notwithstanding the emotional repercussions triggered by the disease process, Costa et al. [5] advocate that full assistance from healthcare professionals may positively influence future expectations of women with breast cancer.

## 4.1. Strengths and limitations of the study

Even though this research was conducted in an important oncological care center in Juiz de Fora, there is another reference hospital in the city, which assists the same region. It is also limited to the macroregion of Zona da Mata in Minas Gerais. On the other hand, the hospital where the research was carried out assists several surrounding cities, covering a very heterogeneous public regarding place of living and age.

Regarding the number of participants, qualitative approach research usually includes a limited number of interviews, ranging from 05 to 28 [3–6, 8, 26, 27] participants, approximately, as the understanding of a phenomenon or an event does not need to be quantified and the depth of that data collection instrument is often sufficient to achieve the proposed goals. In this investigation, due to being part of an ongoing study, 40 interviews were carried out, reflecting the density of the information collected. However, it is admitted that a qualitative research has the limitation of not being representative for other groups and does not propose the generalization of data, since it is designed to answer more specific questions of the object of study [28]. Furthermore, the fact that the researcher who performed data collection was a

woman tended to mitigate any constraints, enabling an identification with the participants and greater freedom to express their feelings and perceptions.

Regarding data collection, this study sought to reflect methodological rigor of a study with a qualitative approach, as all data was collected only by the main researcher, with almost 20 years of training and experience in qualitative research in the field of neoplasm. Analysis of the findings was performed by the main researcher and discussed with the remaining authors, aiming at scrutinizing the meaning of the speech in the interviews and achieve accuracy in the interpretation of findings.

The cross-sectional design was adequate to understand the implications of breast cancer diagnosis from the perspective of the women suffering from the disease. Notwithstanding, such design is limited, as a cross-sectional cut-off prevents the assessment of the studied event in the long run.

Because it is an unexpected event, at the global level, COVID-19 interfered in the data collection of the study, since part of the collection was performed during the pandemic. Therefore, the present study envisioned some repercussions of the health crisis in the process of breast cancer discovery from the perception of women, focus of the study. Because it is a cross-sectional approach to a larger longitudinal research that is in progress, with broader purposes, a directly comparative analysis was not performed regarding the perceptions on the discovery of neoplasm in the pre-pandemic and pandemic periods. It is understood that follow-up studies are needed to verify major repercussions of such event in cases of people suffering from cancer, in the long term, encompassing a larger part of population.

## 5. Conclusion

The diagnostic of breast cancer may be devastating for women suffering from it, and it needs integral and humanized assistance, offered by a multidisciplinary team, to aid in the process of accepting and overcoming the disease, also favoring adherence to the treatment. In such an impacting moment as that of receiving the news, the role of the psychologist is crucial, as the one who will be able to identify and solve specific demands inherent to emotional issues, as well as other matters involving breast cancer. Several women detect some breast changes even before the diagnostic exams. Thus, investment in health education for the population is paramount, as knowledge of the signs and symptoms of the neoplasm may enable women to go to healthcare centers in time for a good prognostic and cure of cancer, mitigating the repercussions in the life of people suffering from it. Family, friends, and spirituality must be taken into consideration by the healthcare team, as an integral part of the supporting network, because they are key factors for women in the process of coping with the disease and the treatments. In that sense, it is important that healthcare professionals are prepared to understand and accept the demands of women in a holistic manner, in the various aspects involving health and illness, feelings, beliefs and values. The COVID-19 pandemic was an obstacle that hampered the tracking, investigation, and diagnostic of breast cancer, however, health services seem to have adapted in order to overcome this situation. The social isolation caused by the pandemic increased the distancing, which interfered in social relations, intensifying feelings of helplessness and anxiety. In that sense, it is worth mentioning the urgent need of social support to women with breast cancer, especially in these times of health crisis, in order to provide the quality assistance, they need, and that future studies may identify greater impact of the pandemic in the long run.

## Acknowledgments

We would like to thank the participants of this study and the researchers who collected the data.

## Author Contributions

**Conceptualization:** Simone Meira Carvalho, Fabiane Rossi dos Santos Grincenkov.

**Data curation:** Simone Meira Carvalho, Fabiane Rossi dos Santos Grincenkov.

**Formal analysis:** Simone Meira Carvalho, Fabiane Rossi dos Santos Grincenkov.

**Funding acquisition:** Simone Meira Carvalho.

**Investigation:** Simone Meira Carvalho.

**Methodology:** Simone Meira Carvalho, Camilla de Abrahão Andrade, Mariana Barbosa Leite Sérgio Ferreira, Karine Soriana Silva de Souza, Fabiane Rossi dos Santos Grincenkov.

**Project administration:** Simone Meira Carvalho.

**Supervision:** Fabiane Rossi dos Santos Grincenkov.

**Validation:** Simone Meira Carvalho.

**Writing – original draft:** Simone Meira Carvalho, Fabiane Rossi dos Santos Grincenkov.

**Writing – review & editing:** Simone Meira Carvalho, Fabiane Rossi dos Santos Grincenkov.

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
