## [Decision Letter · Decision Letter 0]

25 Nov 2022

PONE-D-22-24658Women’s perception about the discovery of breast cancer in the middle of the covid-19 pandemicPLOS ONE

Dear Dr. Carvalho,

Thank you for submitting your manuscript to PLOS ONE. After careful consideration, we feel that it has merit but does not fully meet PLOS ONE’s publication criteria as it currently stands. Therefore, we invite you to submit a revised version of the manuscript that addresses the points raised during the review process.Author(s) need to be specific on the methodology they intend to use to avoid confusing readers. For example, the author(s) uses qualitative methods of data collection (interview) but end-up using the tools (SPSS) for quantitative methods of data analysis.Reviewers differ in decision because of author(s) methodological flaw. To seek for statistical significant you may need to use questionnaire with larger population, although combining methods might enhance the work. If you have available data for such mixed methods, then it will be fantastic to combine both.Author(s) can report part of qualitative data using table(s), but sophisticated analytical tools for qualitative data cannot be utilized to generate such result.

We look forward to receiving your revised manuscript.

Kind regards,

Adetayo Olorunlana, Ph.D.

Academic Editor

PLOS ONE

Journal Requirements:

“No”

Reviewers' comments:

Reviewer's Responses to Questions

**Comments to the Author**

1. Is the manuscript technically sound, and do the data support the conclusions?

Reviewer #1: No

Reviewer #2: No

Reviewer #3: Yes

2. Has the statistical analysis been performed appropriately and rigorously? 

Reviewer #1: No

Reviewer #2: No

Reviewer #3: Yes

3. Have the authors made all data underlying the findings in their manuscript fully available?

Reviewer #1: No

Reviewer #2: No

Reviewer #3: Yes

4. Is the manuscript presented in an intelligible fashion and written in standard English?

Reviewer #1: Yes

Reviewer #2: Yes

Reviewer #3: Yes

5. Review Comments to the Author

Reviewer #1: This is an article about the perception of women with breast cancer during the pandemic. I congratulate the authors for the important initiative, but the impact in this context did not seem clear to me (pre-pandemic versus during the pandemic). A direct comparison between perceptions before and during the pandemic could shed light on this scenario. As a suggestion, objective questions that could be compared between periods could be asked with a larger number of patients.

Reviewer #2: A study on a very interesting aspect of perceiving cancer, in this case breast cancer, in the context of the COVID-19 pandemic. However, the study was conducted on too small a group to draw any more general conclusions. In my opinion, the questionnaire prepared in a vague way did not allow to draw any interesting conclusions. The results section lacks any statistical elaboration to draw more general conclusions regarding breast cancer detection delays during the pandemic. There is also no summary, in my opinion, of what aspects related to the treatment of breast cancer were the most difficult for patients. In the results section, answers from patient surveys are entered.

Reviewer #3: Replace "middle" with "peak" in the topic. It may be relatively difficult to determine the middle of COVID-19 Pandemic.

220: Table 1 (Marital Status should broken into two. 1. a. Ever married b. Never married 2. If ever married, what is the current state of marriage? a. Currently with spouse b. Separated from couples c. Divorced d. Widow. Common law marriage should expunged. The information on the City as provided in the Table 1 should be removed but discussions around it should be in the research methodology. The mix of qualitative and quantitative data enriches this submission.

6. PLOS authors have the option to publish the peer review history of their article (what does this mean?). If published, this will include your full peer review and any attached files.

Reviewer #1: No

Reviewer #2: No

Reviewer #3: **Yes: **Professor Femi Rufus TINUOLA

---

## [Author Response · Author response to Decision Letter 0]

18 Jan 2023

Dear Dr. Carvalho,

Thank you for submitting your manuscript to PLOS ONE. After careful consideration, we feel that it has merit but does not fully meet PLOS ONE’s publication criteria as it currently stands. Therefore, we invite you to submit a revised version of the manuscript that addresses the points raised during the review process.

• Author(s) need to be specific on the methodology they intend to use to avoid confusing readers. For example, the author(s) uses qualitative methods of data collection (interview) but end-up using the tools (SPSS) for quantitative methods of data analysis.

• Reviewers differ in decision because of author(s) methodological flaw. To seek for statistical significant you may need to use questionnaire with larger population, although combining methods might enhance the work. If you have available data for such mixed methods, then it will be fantastic to combine both.

• Author(s) can report part of qualitative data using table(s), but sophisticated analytical tools for qualitative data cannot be utilized to generate such result.

We look forward to receiving your revised manuscript.

Kind regards,

Adetayo Olorunlana, Ph.D.

Academic Editor

PLOS ONE

Journal Requirements:

Answer: The study was carried out upon the receipt of the Free and Informed Consent Form, in written format, signed by all those who agreed to participate in the study (inserted in the Article).

“No”

Answer: There was no funding of any kind. (inserted in the Article)

Answer: There was no funding of any kind; all costs, both from research and from the submission of the article, were subsidized by the main researcher. (inserted in the Article)

Answer: data will not be made available.

Considering that part of the interviews contain information that might identify the participants, the researchers committed themselves to treat the identity of the participants according to professional standards of confidentiality, since they contain particular elements and names of people and/or professionals. Thus, the investigation was based on the Brazilian norms that regulate research with human beings (resolution no. 466, of December 12, 2012, of the National Health Council/Ministry of Health) that uphold the confidentiality of the identification data of the research members, we chose not to provide the complete transcripts. However, speech clippings (transcriptions) were used as a way to represent the findings of the study, seeking to exemplify the categories listed from the content analysis applied.

Reviewers' comments:

Reviewer's Responses to Questions

Comments to the Author

1. Is the manuscript technically sound, and do the data support the conclusions?

Reviewer #1: No

Reviewer #2: No

Reviewer #3: Yes

Answer 1.1: Initially, we the authors would like to thank the observations, questions and suggestions made by the editor and by each reviewer. The scores collaborate with the improvement of this work and enrich the learning for future research and academic productions, contributing with other perspectives to the focus of the research. Thus, we hope to have implemented all the modifications and have complied with every suggestion or request made.

Answer 1.2: This research reflects a cross-sectional approach to a larger study, which is in progress. The proposal was to present a descriptive, exploratory study design with a strictly qualitative approach. Regarding the number of participants, the study had 40 interviews that, based on other studies (see "strengths and limitation of the study"), are sufficient to understand the event studied, that is, the perception of women about the discovery of breast cancer. However, we understand that it is not possible to generalize the findings to other samples, and it is not our intent to be generalizable. In this sense, the conclusions of the research aim to shed some light on the need for an educational approach to identify signs and symptoms that favor screening and early diagnosis of breast cancer. Furthermore, we wish to highlight the importance of preparing the professional team to consider and welcome the emotional impact generated by the diagnosis of a stigmatized disease, considering the support network that women need to cope with the situation. 

2. Has the statistical analysis been performed appropriately and rigorously?

Reviewer #1: No

Reviewer #2: No

Reviewer #3: Yes

Answer 1: This study is strictly qualitative, so no statistical analysis was performed. Only sociodemographic and clinical data were presented in absolute and relative frequency format. The presentation was corrected in the article, to avoid confusion to the reader. (inserted in the Article)

3. Have the authors made all data underlying the findings in their manuscript fully available?

Reviewer #1: No

Reviewer #2: No

Reviewer #3: Yes

Answer: This is a cross-sectional study, with a strictly qualitative approach. Considering that part of the interviews contain information that might identify the participants, the researchers committed themselves to treat the identity of the participants according to professional standards of confidentiality, since they contain particular elements and names of people and/or professionals. Thus, the investigation was based on the Brazilian norms that regulate research with human beings (resolution no. 466, of December 12, 2012, of the National Health Council/Ministry of Health) that uphold the confidentiality of the identification data of the research members, we chose not to provide the complete transcripts. However, speech clippings (transcriptions) were used as a way to represent the findings of the study, seeking to exemplify the categories listed from the content analysis applied.

4. Is the manuscript presented in an intelligible fashion and written in standard English?

Reviewer #1: Yes

Reviewer #2: Yes

Reviewer #3: Yes

Answer: Thank you!

5. Review Comments to the Author

Reviewer #1: This is an article about the perception of women with breast cancer during the pandemic. I congratulate the authors for the important initiative, but the impact in this context did not seem clear to me (pre-pandemic versus during the pandemic). A direct comparison between perceptions before and during the pandemic could shed light on this scenario. As a suggestion, objective questions that could be compared between periods could be asked with a larger number of patients.

Answer: Since this is an unexpected event, at the global level, COVID-19 interfered in the data collection of the study, with part of the collection performed during the pandemic due to the situation. However, this article was a cross-sectional study of a larger longitudinal research that is in the process of completion, and whose main focus is on a health education group intervention for self-care in breast cancer. Thus, a directly comparative analysis was not performed regarding the perceptions about the discovery of neoplasia in the pre-pandemic and pandemic periods.

Reviewer #2: A study on a very interesting aspect of perceiving cancer, in this case breast cancer, in the context of the COVID-19 pandemic. However, the study was conducted on too small a group to draw any more general conclusions. In my opinion, the questionnaire prepared in a vague way did not allow to draw any interesting conclusions. The results section lacks any statistical elaboration to draw more general conclusions regarding breast cancer detection delays during the pandemic. There is also no summary, in my opinion, of what aspects related to the treatment of breast cancer were the most difficult for patients. In the results section, answers from patient surveys are entered.

Answer: This is a cross-sectional study of a longitudinal research, of doctorate level, which is in progress, and had as its main objective the realization of an intervention in a group of health education for self-care in breast cancer, having as a sample women undergoing chemotherapy treatment. Since the pandemic interrupted data collection, there was a need to re-adjust the objectives and the intervention.

 In this cross-sectional, descriptive and qualitative approach, we sought to understand the perceptions of women with breast cancer regarding the diagnosis. The sample of 40 participants, for a qualitative study, has been considered sufficient by other authors to achieve the understanding of an event such as this. It is understood that this approach has limitations regarding the generalization of the findings, which are restricted to the sample addressed in the article. Because breast cancer is a disease of importance in public health, although with limitations, it is intended to alert health professionals to have a broader look on the aspects that involve all the stages of the disease, ever since the discovery of a breast alteration, up until the emotional impact generated by the diagnosis and the possible implications of an unusual situation, as was the COVID-19 pandemic.

 Since the interview script was elaborated and tested for further research, the questions addressed more in-depth questions about self-care during cancer treatments.

However, the question used to start the interview and dive into the theme of breast cancer raised interesting and profound reports on the discovery of the neoplasia, arousing the authors' interest in sharing such information. 

Many women discovered the cancer by themselves, despite having mammographs done, even as often as the Ministry of Health in Brazil recommends for the control and screening of breast neoplasia. They also showed the impact and emotional repercussions of the diagnosis, such as the feelings of helplessness generated by the isolation brought by the pandemic.

 In order to improve the quality of the article and comply with the reviewer's suggestions, some adaptations were made in the "Conclusion", which we hope has reached the expectations of the reviewer. It is noteworthy that, from the perspective of the qualitative approach, the interviewees' perceptions on the theme are considered, assuming the restrictions of this type of study with regard to the generalization and representativeness of the results for a broader population.

 Considering the larger study, the analysis of the material explored (interviews) gave rise to several thematic axes and their categories, including the discovery of breast cancer, beliefs about the disease, perceptions regarding treatments and self-care during the treatment period.

Reviewer #3: Replace "middle" with "peak" in the topic. It may be relatively difficult to determine the middle of COVID-19 Pandemic.

220: Table 1 (Marital Status should broken into two. 1. a. Ever married b. Never married 2. If ever married, what is the current state of marriage? a. Currently with spouse b. Separated from couples c. Divorced d. Widow. Common law marriage should expunged. The information on the City as provided in the Table 1 should be removed but discussions around it should be in the research methodology. The mix of qualitative and quantitative data enriches this submission.

Answer: As suggested by the reviewer, the table was adapted with regard to the city of origin. It has also been corrected with regard to marital status, based on the classification of other studies published in this renowned journal [Della Vecchia C, Préau M, Carpentier C, Viprey M, Haesebaert J, Termoz A, Dima AL, Schott AM. Illness beliefs and emotional responses in mildly disabled stroke survivors: A qualitative study. PLoS One. 2019 Oct 23;14(10):e0223681. doi: 10.1371/journal.pone.0223681.]; [Serafim AP, Durães RSS, Rocca CCA, Gonçalves PD, Saffi F, Cappellozza A, Paulino M, Dumas-Diniz R, Brissos S, Brites R, Alho L, Lotufo-Neto F. Exploratory study on the psychological impact of COVID-19 on the general Population Brazilian. PLoS One. 2021 Feb 3;16(2):e0245868. doi: 10.1371/journal.pone.0245868. PMID: 33534820; PMCID: PMC7857630.]

 This article is a presentation of a cross-sectional approach to a larger longitudinal research, with a main focus on a health education group intervention for self-care in breast cancer. Given the richness and depth of the results obtained in the interviews, several thematic axes emerged that will be explored in other articles.

 As the research is still ongoing, we intended to perform an analysis with mixed methods, which should be presented shortly.

6. PLOS authors have the option to publish the peer review history of their article (what does this mean?). If published, this will include your full peer review and any attached files.

Do you want your identity to be public for this peer review? For information about this choice, including consent withdrawal, please see our Privacy Policy.

Reviewer #1: No

Reviewer #2: No

Reviewer #3: Yes: Professor Femi Rufus TINUOLA

Answer: We, the authors, really appreciate the attention and collaboration made for improvements in the research. Do not hesitate to contact us in case any further clarifications or changes are needed

.

---

## [Decision Letter · Decision Letter 1]

20 Feb 2023

Women’s perception about the discovery of breast cancer in the middle of the covid-19 pandemic

PONE-D-22-24658R1

Dear Dr. Carvalho,

We’re pleased to inform you that your manuscript has been judged scientifically suitable for publication and will be formally accepted for publication once it meets all outstanding technical requirements.

Kind regards,

Adetayo Olorunlana, Ph.D.

Academic Editor

PLOS ONE

Additional Editor Comments (optional):

Reviewers' comments:

Reviewer's Responses to Questions

**Comments to the Author**

1. If the authors have adequately addressed your comments raised in a previous round of review and you feel that this manuscript is now acceptable for publication, you may indicate that here to bypass the “Comments to the Author” section, enter your conflict of interest statement in the “Confidential to Editor” section, and submit your "Accept" recommendation.

Reviewer #2: All comments have been addressed

2. Is the manuscript technically sound, and do the data support the conclusions?

Reviewer #2: Yes

3. Has the statistical analysis been performed appropriately and rigorously? 

Reviewer #2: Yes

4. Have the authors made all data underlying the findings in their manuscript fully available?

Reviewer #2: Yes

5. Is the manuscript presented in an intelligible fashion and written in standard English?

Reviewer #2: Yes

6. Review Comments to the Author

Reviewer #2: I recomend the manuscript "Women’s perception about the discovery of breast cancer in the middle of the covid-19 pandemic" to publication

7. PLOS authors have the option to publish the peer review history of their article (what does this mean?). If published, this will include your full peer review and any attached files.

Reviewer #2: No

---

## [Editor Report · Acceptance letter]

1 Mar 2023

PONE-D-22-24658R1 

Women’s perception about the discovery of breast cancer amid the covid-19 pandemic 

Dear Dr. Carvalho:

I'm pleased to inform you that your manuscript has been deemed suitable for publication in PLOS ONE. Congratulations! Your manuscript is now with our production department. 

Kind regards, 

on behalf of

Associate Professor Adetayo Olorunlana 

Academic Editor

PLOS ONE